# StudentSADD: Mobile Depression and Suicidal Ideation Screening of College Students during the Coronavirus Pandemic

**ML Tlachac, Ricardo Flores, Miranda Reisch, Rimsha Kayastha, Nina Taurich,**
**Veronica Melican, Connor Bruneau, Hunter Caouette, Ermal Toto, Elke Rundensteiner**
Worcester Polytechnic Institute, Worcester, MA 01609
{mltlachac,rflores,mhernandezreisch,rkayastha,ntaurich,
vmelican,cdbruneau,hcaouette,toto,rundenst}@wpi.edu

## Abstract

The growing prevalence of depression and suicidal ideation among college students is alarming, with the Coronavirus pandemic further highlighting the need for universal mental illness screening technology. While traditional screening questionnaires are too burdensome to achieve universal screening in this population, data collected through mobile applications has the potential to identify at-risk students. However, knowing the modalities that students are willing to share and that contain strong screening capabilities is critical for developing such mental illness screening technology. Thus, we deployed a mobile application to over 300 students during the pandemic to collect the Student Suicidal Ideation and Depression Detection (StudentSADD) dataset. Overall, students were most willing to share text responses, unscripted voice recordings, and scripted voice recordings. To provide baselines, we trained machine learning and deep learning methods on these modalities to screen for depression and suicidal ideation. The novel StudentSADD dataset is a valuable resource for developing mobile mental illness screening technologies.

## 1 Introduction

Mental illnesses are very prevalent, especially among college students. According to the national Healthy Minds study (HMS), 39% of the 32 thousand surveyed college students in the U.S. reported experiencing depression in 2020 [1]. When left untreated, depression drastically increases suicide risk [2], disability [3], and developing other life-threatening diseases [4]. 14% of students reported experiencing suicidal ideation in the past year [1]. Suicide is the second leading cause of death for individuals in the 10 to 34 age group in the U.S. [5]. Alarmingly, the percent of U.S. college students with severe depression, suicidal thinking, and self-injury more than doubled in the past decade [6].

The Coronavirus Disease 2019 (COVID-19) pandemic has greatly increased the prevalence of mental illnesses globally. The World Health Organization declared COVID-19 a pandemic on March 11, 2020 and shortly thereafter U.S. states began to issue stay-at-home orders [7]. Between June 2019 and June 2020, the rates of reported depression symptoms in U.S. adults quadrupled to 24.3% and the rates of reported suicidal ideation doubled to 10.7% [8]. These rates were still highest for participants aged 18 to 24; their depression rate was 52.3% and their suicidal ideation rate was 25.5% [9]. While there are many pandemic-related stressors, isolation from social distancing contributes to this startling increase in the prevalence of depression symptoms. Further, 31% of U.S. adults a month into recovering from COVID-19 infection had depression [10], a rate higher than the general public.

While many colleges abruptly transitioned to virtual learning during the Spring 2020 semester, Fall 2020 was the first entire semester impacted by COVID-19 restrictions. According to MHS [11; 1],

Submitted to the 35th Conference on Neural Information Processing Systems (NeurIPS 2021) Track on Datasets and Benchmarks. Do not distribute.

COVID-19 resulted in the percent of students living in college dorms to decrease from $34\%$ to $14\%$ and the percent of students who felt that mental health difficulties frequently hurt their academic performance increased from $20\%$ to $28\%$. Thus, already stressed students were being increasingly socially isolated, further exacerbating the behavior health challenges experienced by these transition age youth. Over $60\%$ of the students reported lack of companionship, feeling left out, and being isolated from others at least some of the time [1]. Even prior to COVID-19, mental health services on many U.S. campuses proved ill-equipped to handle the growing demand of students seeking help [12]. Remote students during the COVID-19 pandemic may not even have access to treatment if they resided in a different state than their school [13]. Given their developing brains, students lacking adequate support are at particular risk of self-medicating through unsafe behaviors.

Therefore, it is crucial to identify at-risk students to connect them with resources. Typically, mental illnesses are screened for with questionnaires [14]. Perceived as intrusive [15], those surveys are unfortunately subject to conscious and unconscious bias. Symptoms of depression may also prevent people from seeking help [16]. Further, students may not address symptoms due to not recognizing them [17] or fear of consequences [13]. Thus, to achieve universal mental illness screening of college students, a more subtle approach is required. Since $95\%$ of college aged adults in the U.S. have smartphones [18], mobile devices may be the ideal conduit for screening this population. Prior studies used smartphone applications (apps) to collect longitudinal sensor data from students for mental health assessment [19; 20; 21; 22]. The ability of voice [23] and social media posts [24; 25] to detect mental illnesses in other populations have also been explored. Previously, Mood Assessment Capable Framework (Moodable) [26] and Early Mental Health Uncovering (EMU) [27] studies collectively assessed the willingness of around $400$ crowd-sourced workers to share smartphone sensor data, audio recordings, and social media posts for depression assessment. However, to date, no such analysis has been conducted on a college student population.

Our current research thus explores the willingness of students to share a wide variety of digital phenotype data and the depression and suicide ideation screening potential of such data. In this work, we present the Student Suicidal Ideation and Depression Detection (StudentSADD) dataset, which we collected through an app (Android and Website) that prompted students to record samples of their voice, share phone and social media data, answer questions, and complete a depression screening survey. Data was collected from over $300$ students throughout the Fall 2020 semester, the first semester to be fully impacted by COVID-19. We use a variety of machine learning methods including pretrained deep learning to analyze the depression and suicidal ideation screening ability of the most shared modalities. Contributions of this dataset and benchmark work include:

1. Presentation of the StudentSADD dataset which contains more students and a richer variety of almost instantaneously obtainable data modalities than prior related collections.

2. Assessment of willingness of students to share a variety of modalities through an app.

3. Evaluation and comparison of the most shared modalities to screen for depression and suicidal ideation with machine learning and pretrained deep learning methods.

## 2 Related literature

There are many mental illness screening survey instruments. The most common for depression screening is the 9-item Patient Health Questionnaire-9 (PHQ-9) [28; 29]. Each item asks users to rank the frequency of a depression symptom from '0: not at all' to '3: almost every day'. An user's PHQ-9 score is the summation of the 9 item scores. The PHQ-9 has a sensitivity and specificity of $88\%$ for depression at the cutoff of 10 [28]. The last item asks about experience with "Thoughts that you would be better off dead, or thoughts of hurting yourself in some way?" When this item-9 regarding suicidal ideation is absent, the survey is referred to as the PHQ-8. The first two questions are referred to as PHQ-2. Alternatives to PHQ include the 16-item Quick Inventory of Depressive Symptomatology (QIDS) [20], the 20-item Center for Epidemiologic Studies Depression Scale (CES-D) [30], and the 7-item depression subscale from the Depression, Anxiety, and Stress Scales (DASS) [21].

Research during the last decade has aimed to to identify alternative screening options that are less biased and intrusive. In particular, social media [24; 25], audio [23], and mobile sensor data [31] have been explored. Rooksby, Morrison, and Murray-Rust [31] interviewed 15 students to determine the acceptability of digital phenotype data being used in mental health surveillance. However, other

research [26] using crowd-sourced workers demonstrated that reported willingness to share modalities does not always correspond to those modalities being shared.

The Moodable [26] and EMU [27] crowd-sourced collection conducted in 2017-2019 are unique in that they concurrently collected audio recordings, social media, and smartphone sensor data through Android apps. The findings from the around 400 crowd-sourced workers indicate that a short scripted audio recording is the most acceptable modality for mobile depression screening [26; 27]. Most of the research that uses audio for depression screening has been conducted on longer voice recordings thanks to the popular Distress Analysis Interview Corpus Wizard-of-Oz (DAIC-WOZ) clinical interview corpus [32; 33] which consists of 189 interviews labeled with PHQ-8 scores. The more recent a Multi-modal Open Dataset for Mental Disorder Analysis (MODMA) dataset [34] contains scripted and unscripted voice recordings from 55 clinically assessed hospital patients with PHQ-9 scores. DiMatteo et al. [35] deployed an Android app to collect two weeks of environmental audio and PHQ-8 scores from 84 crowd-sourced workers in 2019.

However, ease of access makes social media the most common modality for mental illness assessment research. Literature reviews [24; 25] reveal that Twitter is the most popular platform and depression is the most commonly screened for mental illness. For example, De Choudhury et al. [36] collected CES-D scores and one year of tweets from 476 participants who reported being diagnosed with depression to predict the onset of depression. De Choudhury et al. [37] also collected PHQ-9 scores from 165 new mothers to predict depression from Facebook posts. Similarly, Ricard et al. [38] recruited 749 Instagram participants to complete the PHQ-8 to predict depression with Instagram data. While social media posts are similar to text messages, only Tlachac et al. [39] has compared the depression screening ability of these two modalities for over 100 participants with PHQ-9 scores.

While instantaneous mobile mental illness screening is rare [26; 27], traditional *prospective mobile screening apps* are more common. Wang et al. [19] were the first to use continuous smartphone sensing to assess mental health by deploying the StudentLife Android app to 48 students for 10 weeks in 2013. They found the PHQ-9 score was negatively correlated with sleep duration, conversation frequency, and conversation duration [19]. The MoodTraces Android app collected GPS traces and PHQ-8 scores from 28 public users in 2014-2015 so mobility patterns could be analyzed as a modality to assess depressive mood disorders [40]. The LifeRhythm app [20] was deployed twice in 2015-2017 to collectively 183 students, 79 of whom completed the PHQ-9 and 104 of whom completed QIDS. Support vector classifiers were trained on features extracted from six weeks of Wifi data to detect depression symptoms for these students [20]. The DemonicSalmon study [21] deployed an Android app in 2016 to 72 students to identify the manifestation of depression and anxiety symptoms in two weeks of prospective smartphone sensor data.

Most recently, the StudentLife Android app [22] was deployed again in early 2020 for 6 weeks to 178 Dartmouth College students to determine if COVID-19 news was associated with higher PHQ-2 scores. Throughout the six week winter term (Jan 5 - March 13), the students phone usage increased, physical activity decreased, and locations visited decreased based on their smartphone sensor data [22]. However, it is unclear if this was due to the COVID-19 news or the continuation of an existing trend. Most universities on the East Coast were not directly impacted until after Dartmouth College's winter term ended on March 13. While we also use an app to collect student data for mental health assessment, our research differs in a number of key ways. First, the purpose of our research is to determine student willingness to share different modalities. Second, we aim to assess the ability of different modalities to develop almost instantaneous screening technologies. As such, our app collects a wide range of modalities during a single quick session. Lastly, we collect data when students were either virtual or faced severe COVID-19 restrictions on campus.

# 3 Data collection and machine learning methodology

We collected the Student Suicidal Ideation and Depression Detection (StudentSADD) dataset from students between August 2020 and January 2021 under WPI IRB 00007374 File 18-0031. We began collecting data in the month prior to WPI's first full semester impacted by COVID-19 and ended it the month after the semester concluded. We deployed an Android app and a web app to collect data from undergraduate and graduate students. These apps administered the popular PHQ-9 screening instrument to provide depression and suicidal ideation labels for the data.

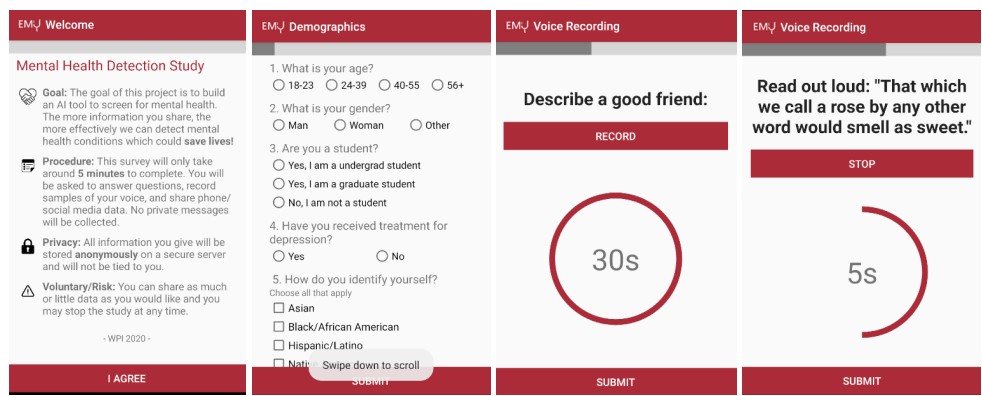

Figure 1: Four pages from the StudentSADD mobile collection app. The first page displays the IRB-approved instructions that describes the goal, procedure, and risk.

### 3.1 The StudentSADD collection applications

We modified the EMU Android app [27] to collect PHQ-9 scores, demographics, retrospective phone logs, text prompt, voice recordings, location history, and tweets. We estimated that sharing all modalities would take at most five minutes. The progression of selected app pages, displayed in Figure 1, were informed by prior research [26; 27] to maximize data quantity. As not all students have Android phones, we also developed an abbreviated web app that collects compatible modalities. The web app collected PHQ-9 scores, demographics, text prompt, voice recordings, and tweets. Assuming many students would complete the web app on their smartphones, we designed it to be accessible on common mobile devices. All collected data was TSL encrypted and sent to our secure server.

The original five demographic questions are displayed in Figure 1. While we placed the demographic page at the end of the collection during our initial trial deployment, we moved it directly after the PHQ-9 page when we noticed not all participants reached it. Given the impact of COVID-19 on mental health [8], we also added two COVID-19 related questions to the demographics page part way through the semester. The first question is designed to gauge social isolation: *Have you been working/studying remotely?* The second question is more direct: *Have you had COVID-19?*

The Android app next asked for permission to collect text logs, call logs, contacts, and calendar entries stored on the phone. We collected the text logs without content to preserve privacy. All phone modalities were optional. Participants were asked to give individual permission for each modality shared to guarantee informed consent. To further preserve privacy, we performed a one-way hash function on all numbers and names in structured data fields prior to sending data to our secure server.

To collect self-written text from the majority of participants, we included a text prompt which we hypothesized (and tested) all students would be willing to share. Specifically, we prompted participants to "describe your favorite place" in under 2000 characters. As the unscripted audio prompt was similar to the text prompt, we placed it next to help users understand the type of response to record. The unscripted audio prompt,"describe a good friend", was chosen to be intentionally vague to elicit a variety of interpretations. For scripted audio, the apps prompted participants to read an iconic Shakespeare quote. Participants had 30 and 10 seconds respectively to record these prompts, though they could stop recording and submit once half of the allotted time had passed.

Participants with the Android app could share their GPS data stored by their Google account. Both apps asked for Twitter usernames to collect publicly available tweets. Participants could indicate they did not have a Twitter account or simply decline to share. As with the phone data, we performed a one-way hash functions on all structured data fields with identifiable information. Upon completion, the app presented links to the national suicide prevention lifeline and a form to contact study staff.

### 3.2 Participant recruitment and incentives for participation

Participant inclusion criteria involved being a postsecondary student and at least 18 years of age. During our summer development phase, we used snowball sampling to recruit students at multiple

universities with different mobile devices. After that, we sent calls for participation to students in a variety of email lists including research groups, classes, and clubs. While most emails were sent to WPI students, we did not restrict our participant population to a single postsecondary institution. We also posted calls for participation to student pages on social media sites.

Participation was voluntary. To motivate data sharing, we implemented a raffle system in which every shared modality was rewarded with a raffle ticket. Students could elect to enter the raffle by sharing their student email. For every valid student email, we allocated one dollar to the raffle to be awarded at the start of the fall semester, end of the fall semester, and start of the spring semester at WPI. This raffle was advertised with $25 increments prior to the fall semester and $50 increments once the fall semester began. We reached the required raffle entries to pay one $25 award and three $50 awards. Further, some professors may have offered their students nominal extra credit for participating.

### 3.3   Data description and cleaning

During the six months of our collection, 302 students submitted 345 sessions. As almost every students submitted a text prompt, we were able to use the text content to help identify sessions completed by the same students. In most cases, these repetitive responses were completed subsequently, indicating that a participant must have exited and restarted the survey - possibly due to technical difficulties. In other cases, we suspect the student responded to multiple calls for participation throughout the semester. Some participants informed us they were unable to submit audio. While we updated the instructions on the call for participation and modified the apps accordingly, the number of participants who submitted audio thus represents the lower bound for willingness to share. Further, not all of the audio recordings contained voice, though in some cases this is due to poor audio quality rather than unwillingness to share voice. Even after restricting our set to good quality voice recordings, voice recordings were still the second most plentiful modality after text. We transcribed voice recordings with Speech Recognition [41]. We replaced some proper nouns in the transcripts and text prompts to protect participant privacy. An example of the data submitted and released is available in Table 1.

### 3.4   Machine learning methodology

To provide baselines, we train a variety of machine learning models to screen for depression (PHQ$\geq$ 10) and suicidal ideation (item9$\geq$ 1) on features and feature embeddings extracted from the most shared StudentSADD modalities: text prompt, unscripted voice recording, scripted voice recording. We consider both audio and transcript in the unscripted voice recording as screening modalities. For the baselines, we only use a single data session from each participant. The deep learning models were run on an internal cluster using NVIDIA K80 NVIDIA V100, and NVIDIA T4 GPUs.

**Feature engineering.** Similar to prior feature engineering protocols applied on text messages [39] and voice transcripts [42], we extract 36 part of speech (POS) tags with TextBlob [43] and 194 lexical word category frequencies with Empath [44] from the text replies and unscripted voice transcripts. We also extract the number of characters and words in these text modalities. Due to the short responses, the feature matrices were sparse. Similar to prior research using audio to detect depression [26; 45], we extract 2268 openSMILE [46] features from each voice recording as defined by AVEC 2013 [47].

**Feature reduction.** We normalize the training data before applying both principal component analysis (PCA) and chi-squared feature selection to reduce the number of features [48]. We train models with up to ten principal components and up to ten chi-squared selected features. The top ten principal components are those that explain the most variance in the features. The chi-squared statistic is calculated between the features and target variable to find the top ten chi-squared features.

**Traditional Machine learning algorithms.** After initial exploration of methods and parameters [48], we screen for mental illnesses with methods including support vector classifier (SVC) with Gaussian kernel, logistic regression with L1 regularization, and k-Nearest Neighbor (kNN) with three neighbors. We also experiment with two tree-based ensemble methods: random forest [48] and eXtreme Gradient Boosting (XGBoost) [49]. For both we set the maximum depth of the trees to three to prevent overfitting. These algorithms were trained with the aforementioned text and audio features.

**Deep learning with text.** For the two text modalities, we use Bidirectional Encoder Representations from Transformers (BERT) [50], a state-of-the-art model for NLP tasks, to create text feature embeddings. BERT uses Transformers [51], and is pre-trained over two tasks, predicting missing

Table 1: *Examples of modalities submitted by student participants through the collection apps compared to the data that is shared as part of the StudentSADD dataset upon paper release. The same types of data were collected and released for scripted audio as for unscripted audio. As StudentSADD is not a static dataset, more feature sets may become available, especially for the audio modalities.*

| Modality | Participant | Data submitted | Data shared |
|---|---|---|---|
| Text Prompt | 1607315333 (web app) | "Savannah, Georgia\nTrees everywhere, old charm, music, history" | "[City], [State]\nTrees everywhere, old charm, music, history", POS and lexical category text features |
| Unscripted Audio | 4549 (phone app) | 3gp encoding | "someone I can be with and be myself", 2268 openSMILE features, shareAPrompt = "Yes" |
| Twitter | 6831 (phone app) | one-way hashed shared Twitter username | hasTwitter = "Yes", shareUsername = "Yes" |
| GPS | 6831 (phone app) | 84 location logs without location details | shareGPS = "Yes" |
| Calendar | 3517 (phone app) | 12 calendar entries without event information | collected calendar logs, shareCalendar = "Yes" |
| Contacts | 3517 (phone app) | 430 pairs of one-way hashed names and phone numbers | collected contact logs, shareContacts = "Yes" |
| Call Logs | 3517 (phone app) | 1750 call logs with one-way hashed phone numbers | collected call logs, shareCLog = "Yes" |
| Text Logs | 3517 (phone app) | 1734 text logs with one-way hashed phone numbers and no message content | collected text logs, shareTLog = "Yes" |

words and predicting the next sentence. Specifically, we use pretrained BERT as feature-embedding model, and add a classification layer on top of transformer output. We also experiment with two variation of BERT: BERT-LSTM and BERT-Attention. BERT-LSTM includes a Long Short-Term Memory layer over the transformer output [52]. To capture the relationships between longer text, we add self-attention [53] on top of the BERT-LSTM, which we then called the BERT-Attention model. For the implementation of all three of the aforementioned BERT models, we use cross entropy loss function, Adam optimizer, $2e^{-5}$ for learning rate, a step size of $2e^{-8}$, and 128 for maximum number of tokens. We fine-tune these models for each of our tasks.

**Deep learning with audio.** For the voice recordings, we use the popular pretrained audio architecture VGGish [54] to create audio feature embeddings. VGGish transforms voice clips to log Mel spectrograms that are processed by a multilayer convolutional network to extract embeddings vector of size 128 for each second of voice, forming a 2D array that can be used for classification. Like BERT-attention, we add self-attention over the embeddings of VGGish.

**Evaluation.** We designate a stratified sample *test set* for StudentSADD to ensure the training and test sets have similar distributions of binary depression screening scores, binary suicidal ideation screening scores, and quantity of students who shared audio. We upsample the training set with the same random seed (42) prior to training the models. To evaluate the screening ability of each model configuration, we repeat each experiment 10 times and report on the average and standard deviation of the accuracy and F1 scores of the models. The metrics are calculated in Eq. 1 with the number of true positive ($TP$), false positive ($FP$), false negative ($FN$), and true negative ($TN$) predictions.

$$Accuracy = \frac{TP + TN}{TP + FP + FN + TN}, \quad F1 = \frac{2TP}{2TP + FP + FN} \tag{1}$$

## 4 Results

### 4.1 Description of StudentSADD participants and data

Of the 302 students in the StudentSADD dataset, almost half (47.0%) screened positive for depression (PHQ-9$\geq$ 10) and just over a quarter (26.5%) reported suicidal ideation. These rates are higher

Table 2: *The count of students who fall into each group and percent of total population. Average PHQ-9 and item-9 scores $\pm$ standard deviation and count of students who screened positive for depression (PHQ-9 $\geq 10$) and suicidal ideation (item-9 $\geq 1$) are also reported for each group. The percent who screened positive is calculated from the students who are part of that group. While selected as part of multiple groups, no participant selected only 'Native Islander/Pacific Islander'. The one participant who preferred not to identify race/ethnicity had a PHQ-9 score of $0$.*

|  | Total | PHQ-9 | Depressed | Item-9 | Ideation |
|---|---|---|---|---|---|
| Website | 269 (89.1%) | 10.29 ± 6.54 | 128 (47.6%) | 0.45 ± 0.85 | 75 (27.9%) |
| Android | 33 (10.9%) | 9.03 ± 6.13 | 14 (42.4%) | 0.27 ± 0.75 | 5 (15.2%) |
| Age: 18 − 23 | 240 (81.4%) | 10.38 ± 6.59 | 118 (49.2%) | 0.47 ± 0.87 | 67 (27.9%) |
| Age: 24 − 39 | 52 (17.6%) | 8.81 ± 6.18 | 18 (34.6%) | 0.31 ± 0.69 | 11 (21.2%) |
| Age: 40 − 55 | 3 (1.0%) | 16.33 ± 3.77 | 3 (100.0%) | 0.67 ± 0.94 | 1 (33.3%) |
| Woman | 174 (59.0%) | 10.57 ± 6.22 | 86 (49.4%) | 0.43 ± 0.85 | 43 (24.7%) |
| Man | 108 (36.6%) | 8.98 ± 6.96 | 42 (38.9%) | 0.43 ± 0.84 | 28 (25.9%) |
| Other Gender | 13 (4.4%) | 14.38 ± 4.94 | 11 (84.6%) | 0.77 ± 0.07 | 8 (61.5%) |
| Undergrad | 236 (80.0%) | 10.46 ± 6.56 | 117 (49.6%) | 0.48 ± 0.88 | 68 (28.8%) |
| Grad | 59 (20.0%) | 8.95 ± 6.42 | 22 (37.3%) | 0.29 ± 0.69 | 11 (18.6%) |
| No Treatment | 217 (73.6%) | 9.45 ± 6.39 | 90 (41.5%) | 0.40 ± 0.85 | 50 (23.0%) |
| Prior Treatment | 78 (26.4%) | 12.14 ± 6.60 | 49 (62.8%) | 0.55 ± 0.83 | 29 (37.2%) |
| White | 186 (63.1%) | 9.99 ± 6.04 | 87 (46.8%) | 0.37 ± 0.72 | 48 (25.8%) |
| Asian | 59 (20.0%) | 9.47 ± 7.41 | 25 (42.4%) | 0.58 ± 1.06 | 16 (27.1%) |
| Hispanic/Latino | 9 (3.1%) | 9.22 ± 5.88 | 5 (55.6%) | 0.11 ± 0.31 | 1 (11.1%) |
| Black | 10 (3.4%) | 10.00 ± 6.39 | 4 (40.0%) | 0.20 ± 0.40 | 2 (20.0%) |
| Other | 10 (3.4%) | 14.30 ± 7.20 | 7 (70.0%) | 0.80 ± 0.98 | 5 (50.0%) |
| Multiple Groups | 20 (6.8%) | 12.65 ± 7.00 | 11 (55.0%) | 0.80 ± 1.21 | 7 (35.0%) |
| Remote | 97 (53.9%) | 9.90 ± 6.86 | 39 (40.2%) | 0.47 ± 0.87 | 27 (27.8%) |
| Hybrid | 73 (40.6%) | 11.16 ± 6.91 | 38 (52.1%) | 0.62 ± 0.97 | 26 (35.6%) |
| Not Remote | 10 (5.6%) | 14.10 ± 7.53 | 7 (70.0%) | 0.80 ± 0.98 | 5 (50.0%) |
| COVID-19 | 12 (6.3%) | 8.58 ± 5.71 | 4 (33.3%) | 0.25 ± 0.43 | 3 (25.0%) |
| No/Unknown | 168 (88.4%) | 10.79 ± 7.06 | 80 (47.6%) | 0.57 ± 0.95 | 55 (32.7%) |

Table 3: *The count of students who shared each modality. Average PHQ-9 and item-9 scores $\pm$ standard deviation and the count of students who screened positive for depression (PHQ-9 $\geq 10$) and suicidal ideation (item-9 $\geq 1$) are reported. The percent who screened positive is calculated from the students who shared that modality. The percent who shared each modality is calculated from the students who could have shared that modality. For example, only $171$ students reached the Twitter page. Further, only the $33$ students who used the Android app could share GPS and phone modalities.*

|  | Shared | PHQ-9 | Depressed | Item-9 | Ideation |
|---|---|---|---|---|---|
| All | 302 (100.0%) | 10.15 ± 6.51 | 142 (47.0%) | 0.43 ± 0.84 | 80 (26.5%) |
| Demographics | 295 (97.7%) | 10.16 ± 6.56 | 139 (47.1%) | 0.44 ± 0.85 | 79 (26.8%) |
| Text Prompt | 298 (98.7%) | 10.22 ± 6.50 | 141 (47.3%) | 0.44 ± 0.84 | 80 (26.9%) |
| Unscripted Audio | 200 (66.2%) | 9.77 ± 6.25 | 90 (45.0%) | 0.36 ± 0.78 | 44 (22.0%) |
| Scripted Audio | 194 (64.2%) | 9.87 ± 6.32 | 89 (45.9%) | 0.37 ± 0.79 | 44 (22.7%) |
| Unscripted Voice | 110 (55.0%) | 9.51 ± 6.26 | 44 (40.0%) | 0.30 ± 0.68 | 22 (20.0%) |
| Scripted Voice | 115 (59.3%) | 9.43 ± 6.27 | 45 (39.1%) | 0.29 ± 0.67 | 22 (19.1%) |
| Has Twitter | 47 (27.5%) | 10.28 ± 6.25 | 21 (44.7%) | 0.43 ± 0.79 | 13 (27.7%) |
| Username | 16 (34.0%) | 8.31 ± 4.27 | 5 (31.3%) | 0.38 ± 0.78 | 4 (25.0%) |
| GPS | 21 (63.6%) | 7.43 ± 4.11 | 7 (33.3%) | 0.10 ± 0.29 | 2 (9.5%) |
| Calendar | 11 (33.3%) | 7.18 ± 4.32 | 5 (45.5%) | 0.0 ± 0.0 | 0 (0.0%) |
| Contacts | 11 (33.3%) | 7.18 ± 4.32 | 5 (45.5%) | 0.0 ± 0.0 | 0 (0.0%) |
| Call Logs | 10 (30.3%) | 7.70 ± 4.20 | 5 (50.0%) | 0.0 ± 0.0 | 0 (0.0%) |
| Text Logs | 10 (30.3%) | 7.70 ± 4.20 | 5 (50.0%) | 0.0 ± 0.0 | 0 (0.0%) |

than those reported by the HMS study [1] but are similar to the rates for this age group in a more generalized survey [8]. We suspect that selection bias contributed to the higher rates of depression in StudentSADD when compared to the HMS study [1], especially as some of the student groups who distributed our call for participation were mental health focused. Only 33 students shared data through the mobile app and 269 shared data through the web app. Thus, the students demonstrated a distinct preference for sharing data for mental illness screening through a website app.

We have demographics for 295 of the 302 students in the StudentSADD dataset, displayed in Table 2. After moving the demographic page directly after the page of the app that administered the PHQ-9, all participants completed the demographic questions. The disparity between the number of undergraduate and graduate students can be explained by the collection being dispersed through more undergraduate mailing lists. Overall, more younger undergraduate students reported experiencing depression symptoms and suicidal ideation. While the women reported experiencing more severe depressive symptoms than the men, the other gender identified individuals reported much higher average PHQ-9 and item-9 scores. The students who identified with other or multiple racial/ethnic groups also reported the higher average PHQ-9 and item-9 scores than other groups. The participants who identified as only Hispanic/Latino reported the lowest average PHQ-9 and item-9 scores of the racial/ethnic groups.

180 and 190 students responded to the first and second COVID-19 related questions, respectively. Students who were not remote reported the highest average PHQ-9 scores and students who were completely virtual reported the lowest average PHQ-9 scores. While our attempt was to capture social isolation, it is possible that we instead captured privilege or family support. Only 12 participants reported having had COVID-19. These individuals had an average PHQ-9 score of 8.58 and an average item-9 score of 0.25, which is surprisingly lower than all 190 students who answered this question. We hypothesize the students in our study who had COVID-19 were more social.

98.7% of students shared the text prompt, making it the most shared modality. The text prompts ranged between 1 and 355 words. In addition to text prompt, Table 3 shows the willingness of participants to share Twitter and phone related modalities. Additionally, the table displays the number of participants who shared audio. Though the later values may not reflective of willingness to share as some participants contacted us expressing inability to record audio. Further, some of the audio samples did not contain voice or were of poor quality. So, we also report the number of audio recordings that yielded transcripts. Despite these challenges, we observe that the average PHQ-9 scores of students with audio recordings is lower than that of all students. Students who shared phone modalities, GPS, and Twitter username had noticeably lower PHQ-9 scores than all students. None of the students who shared the four phone log modalities reported experiencing suicidal ideation.

## 4.2 Screening results on StudentSADD text and audio

When screening for depression with text, the highest performing models (Table 4) only used one feature. For the text prompt, this feature was the frequency of words in the category 'optimism'. However, the highest accuracy and F1 scores for these models were 0.57 and 0.67 respectively. Thus, while more features may be helpful, those were not captured in our feature set. The BERT models had similar F1 scores but higher accuracies, making them more successful at screening for depression with text. For the unscripted transcript, the single feature used by the models was the first principal component. As displayed in Table 4, the accuracy of these models was higher than the text models, but F1 score was lower. This indicates the machine learning models did not have many true positive predictions. BERT models in comparison had lower accuracies but higher F1 scores. Screening for suicidal ideation with text features proved to be a more challenging task given the models used more features but resulted in lower F1 scores. The BERT models also proved more successful at this task.

For unscripted audio, the ensemble methods were the most successful at screening for depression. The highest performing models for this task only used one openSMILE feature: 'F0final_sma_upleveltime50'. When screening for depression, the XGBoost models that use unscripted audio performed similarly to the VGGish with attention models that use scripted audio as observed in Table 5. For both types of audio, VGGish was best for screening for suicidal ideation. The VGGish model trained on unscripted voice recordings was more successful at screening for suicidal ideation than any other models in Tables 4 and 5. However, as evidenced by the higher F1 scores in Table 4, the BERT models were able to identify more depressed students with text prompts than VGGish models with scripted audio.

Table 4: *Machine Learning Text and Transcript Results: average ± standard deviation of accuracy and F1 scores for the highest performing model configurations. The highest performing depression screening models used only one chi-squared selected feature for text and one principal component for transcripts. The highest performing suicidal ideation screening models used less than eight principal components (with the exception of poorly performing Gaussian SVC with text).*

| Method | Data | Depression | | Suicidal Ideation | |
| --- | --- | --- | --- | --- | --- |
| | | Accuracy | F1 | Accuracy | F1 |
| Gaussian SVC | Text | $0.52 \pm 0.00$ | $0.66 \pm 0.00$ | $0.48 \pm 0.00$ | $0.38 \pm 0.00$ |
| Logistic Regression | Text | $0.57 \pm 0.00$ | $0.65 \pm 0.00$ | $0.55 \pm 0.00$ | $0.37 \pm 0.00$ |
| kNN | Text | $0.52 \pm 0.00$ | $0.64 \pm 0.00$ | $0.65 \pm 0.00$ | $0.45 \pm 0.00$ |
| Random Forest | Text | $0.55 \pm 0.01$ | $0.66 \pm 0.01$ | $0.58 \pm 0.03$ | $0.41 \pm 0.03$ |
| XGBoost | Text | $0.57 \pm 0.00$ | $0.67 \pm 0.00$ | $0.67 \pm 0.00$ | $0.23 \pm 0.00$ |
| BERT | Text | $0.64 \pm 0.02$ | $0.65 \pm 0.01$ | $0.72 \pm 0.00$ | $0.45 \pm 0.01$ |
| BERT-LSTM | Text | $0.64 \pm 0.02$ | $0.65 \pm 0.01$ | $0.69 \pm 0.03$ | $0.45 \pm 0.01$ |
| BERT Attention | Text | $0.63 \pm 0.01$ | $0.67 \pm 0.01$ | $0.66 \pm 0.01$ | $0.39 \pm 0.02$ |
| Gaussian SVC | Transcript | $0.48 \pm 0.00$ | $0.41 \pm 0.00$ | $0.67 \pm 0.00$ | $0.35 \pm 0.00$ |
| Logistic Regression | Transcript | $0.52 \pm 0.00$ | $0.47 \pm 0.00$ | $0.64 \pm 0.00$ | $0.14 \pm 0.00$ |
| kNN | transcript | $0.55 \pm 0.00$ | $0.35 \pm 0.00$ | $0.55 \pm 0.00$ | $0.21 \pm 0.00$ |
| Random Forest | Transcript | $0.70 \pm 0.03$ | $0.35 \pm 0.06$ | $0.65 \pm 0.02$ | $0.15 \pm 0.01$ |
| XGBoost | Transcript | $0.67 \pm 0.00$ | $0.42 \pm 0.00$ | $0.64 \pm 0.00$ | $0.14 \pm 0.00$ |
| BERT | Transcript | $0.56 \pm 0.01$ | $0.63 \pm 0.01$ | $0.75 \pm 0.00$ | $0.47 \pm 0.00$ |
| BERT-LSTM | Transcript | $0.57 \pm 0.01$ | $0.64 \pm 0.00$ | $0.75 \pm 0.00$ | $0.46 \pm 0.00$ |
| BERT Attention | Transcript | $0.55 \pm 0.00$ | $0.45 \pm 0.17$ | $0.74 \pm 0.01$ | $0.46 \pm 0.00$ |

Table 5: *Machine Learning Audio Results: average ± standard deviation of accuracy and F1 scores for the highest performing model configurations. For unscripted audio, the highest performing ensemble methods only used one chi-squared selected feature when screening for depression and one principal component when screening for suicidal ideation. For scripted audio, the traditional machine learning and ensemble methods all performed best when using principal components.*

| Method | Audio | Depression | | Suicidal Ideation | |
| --- | --- | --- | --- | --- | --- |
| | | Accuracy | F1 | Accuracy | F1 |
| Gaussian SVC | Unscripted | $0.55 \pm 0.00$ | $0.44 \pm 0.00$ | $0.70 \pm 0.00$ | $0.29 \pm 0.00$ |
| Logistic Regression | Unscripted | $0.55 \pm 0.00$ | $0.48 \pm 0.00$ | $0.67 \pm 0.00$ | $0.27 \pm 0.00$ |
| kNN | Unscripted | $0.64 \pm 0.00$ | $0.54 \pm 0.00$ | $0.64 \pm 0.00$ | $0.33 \pm 0.00$ |
| Random Forest | Unscripted | $0.73 \pm 0.02$ | $0.51 \pm 0.04$ | $0.66 \pm 0.02$ | $0.39 \pm 0.00$ |
| XGBoost | Unscripted | $0.73 \pm 0.00$ | $0.57 \pm 0.00$ | $0.79 \pm 0.00$ | $0.46 \pm 0.00$ |
| VGGish | Unscripted | $0.68 \pm 0.02$ | $0.51 \pm 0.01$ | $0.83 \pm 0.03$ | $0.56 \pm 0.06$ |
| VGGish Attention | Unscripted | $0.67 \pm 0.00$ | $0.51 \pm 0.10$ | $0.81 \pm 0.00$ | $0.37 \pm 0.01$ |
| Gaussian SVC | Scripted | $0.53 \pm 0.00$ | $0.47 \pm 0.00$ | $0.74 \pm 0.00$ | $0.40 \pm 0.00$ |
| Logistic Regression | Scripted | $0.65 \pm 0.00$ | $0.50 \pm 0.00$ | $0.65 \pm 0.00$ | $0.33 \pm 0.00$ |
| kNN | Scripted | $0.56 \pm 0.00$ | $0.52 \pm 0.00$ | $0.76 \pm 0.00$ | $0.50 \pm 0.00$ |
| Random Forest | Scripted | $0.58 \pm 0.02$ | $0.44 \pm 0.04$ | $0.69 \pm 0.02$ | $0.37 \pm 0.02$ |
| XGBoost | Scripted | $0.62 \pm 0.00$ | $0.38 \pm 0.00$ | $0.76 \pm 0.00$ | $0.43 \pm 0.00$ |
| VGGish | Scripted | $0.69 \pm 0.02$ | $0.56 \pm 0.06$ | $0.82 \pm 0.00$ | $0.43 \pm 0.00$ |
| VGGish Attention | Scripted | $0.75 \pm 0.02$ | $0.57 \pm 0.01$ | $0.83 \pm 0.02$ | $0.31 \pm 0.08$ |

# 5   Discussion

## 5.1   Data and software availability

Upon publication, other researchers may access our data analysis code and apply for access to the anonymized StudentSADD dataset at our project website: emutivo.wpi.edu. We will share features and embeddings for the data that can not be anonymized, as noted in Table 1. We include data for all $345$ sessions as the repeated sessions may still have use for data balancing or data generation. We will also share the detailed results for the machine learning models in this paper. Further, this is not a static dataset and we will continue to add more features and embedding representations.

## 5.2   Intended use of StudentSADD data

The data and machine learning baselines can be used by academics to inform the development of digital mental illness screening technologies that could be deployed more universally than traditional screening surveys instruments and connect at-risk individuals with resources. Multiple ways in which the resources in this paper could be used to further the goal of developing such screening technologies exist. The data could be used to train machine learning models that can screen for mental illnesses. To facilitate this objective, we have provided depression and suicidal ideation screening baselines with a specific test set for comparison purposes. Further, the findings regarding what modalities students are willing to share and the ability of these modalities to screen for mental illnesses could inform the design of screening technologies as well as the design of future data collections.

## 5.3   Societal impacts and ethical considerations

Short voice recordings and text are easy to collect. While this makes them great modalities for screening technologies, these modalities could also be collected without the knowledge of the individual who produced the data. Thus, a screening technology created from such modalities could be used to discriminate against individuals with mental illnesses without their knowledge. However, the ethical implications of bad actors would remain regardless of the release of the StudentSADD dataset. Notably, the DAIC-WOZ [32] clinical interview audio and transcripts are already publicly available. Further, publicly available social media posts have been widely used by other mental illness screening research [24; 25]. Therefore, the release of the StudentSADD dataset to help develop screening technology that can connect at-risk individuals with resources outweighs the risk of misuse, especially given the increasing depression rates in students [6] and the general population [8]. There is also evidence that among college students mental illness stigma is decreasing; only $6\%$ of students surveyed by HMS in Fall 2020 would think less of someone for seeking mental illness treatment [1].

## 5.4   Limitations

Our student participants showed a distinct preference for completing the data collection through a website rather than downloading an app. This resulted in a small sample size to determine the willingness of students to share phone modalities for mental illness screening purposes. Further, as the website could be accessed by many different devices with different recording capabilities, not all participants were able to record and share usable audio. Thus, while were unable to determine the exact percent of students who were willing to share audio recordings, we were able to capture relative willingness to share, which can be leveraged by future research. Further, while the design of our app was informed by prior research [26; 27], page order may have had an impact on data shared.

# 6   Conclusion

The 302 students in StudentSADD showed a preference for sharing data through the website app instead of the phone app. Text responses, unscripted voice recordings, and scripted voice recordings were the most shared modalities. In our baseline models, BERT was able to screen for depression with the text responses with an accuracy of $0.64$ and F1 of $0.65$. For suicidal ideation screening, VGGish was able able to achieve an accuracy of $0.83$ and F1 of $0.56$. Collected during the COVID-19 pandemic, our StudentSADD dataset is a valuable resource for developing unobtrusive technologies that can provide universal mental illness screening to at-risk populations.

## Acknowledgements

This work was financially supported by the US Department of Education P200A150306: GAANN grants and the WPI data science department. We thank Ada Dogrucu, Alex Perucic, Anabella Isaro, Damon Ball, and Prof Emmanuel Agu at WPI for innovating the instantaneous mobile screening approach. We thank Professors Fatemeh Emdad, Lane Harrison, Chun-Kit (Ben) Ngan, Peter Hart-Brinson, and everyone else who distributed our call for participation. We thank Bumper the Border Collie, Joshua Lovering, prior Emutivo student teams, and the DAISY lab at WPI for their support.

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

# A   Appendix

## A.1   StudentSADD data description

The Student Suicidal Ideation and Depression Detection (StudentSADD) dataset includes 345 sessions of data from 302 unique student participants with modalities including participant demographics, PHQ-9 scores, responses to a text prompt, unscripted audio recordings, scripted audio recordings, Twitter data (whether or not the participant has Twitter and their Twitter username), and retrospective phone data (calendar, call logs, text logs, contacts and GPS). Participants were given access to an Android app and Website Survey. They could choose which version of the survey took. All questions were the same, however, phone data was only collected from students who used the Android app. All publicly available data is stored in CSV files.

Responses to the text prompt, demographic questions, phq-9 questions, willingness to share, and timestamps of submission by participants, can be found in summaryDataStudentSADD.csv. Participants were asked to share their favorite place (text prompt); demographic info, specifically, age, gender, student status, race/ethnicity, whether or not they have had prior depression treatment, whether they were remote, and whether or not they had covid; responses to 9 depression screening questions with four point Likert scales; GPS data (if shared); and Twitter information, such as whether or not they have Twitter and their username. Participant willingness to share audio and phone data was also included in this file. The sessions from the same student are also marked as copies in this CSV file.

The file phoneDataStudentSADD.csv includes data collected from those participants who used the Android app and elected to share phone data. Each phone modality could be shared or denied. Phone data modalities that could be elected to be shared were calendar entries, call logs, text logs, and contact entries. All names and phone numbers in these data modalities were one-way hashed. The file includes calendar and contacts entries, for which the count can be extracted. The call and text logs include a one-way hashed address (phone number) of the sender/receiver, as well as the timestamp and size/length of each correspondence.

The file scriptedTranscript.csv contains scripted audio transcripts and the file unscriptedTranscript.csv contains unscripted audio transcripts. These CSV files also have the PHQ-9 and item-9 scores of the corresponding participants. Feature and embedding representations of the audio data will be released instead of the raw audio data to protect student privacy. The files scriptedSMILE.csv and unscriptedSMILE.csv contain examples of extracted audio features.

Further, detailed results for machine learning models that use the StudentSADD data will be released. The files baselineScripedML.csv, baselineUnscriptedML.csv, baselineTranscriptML.csv, and baselineTextML.csv contain examples of detailed machine learning model results.

84 IDs were designated as a test set to evaluate the machine learning models trained on this data. The Test and Train IDs can be found below. In addition to the training IDs from unique students, we have also included the IDs from duplicated entries. Note, not all IDs shared every modality.

Test IDs: [1607712777, 292, 2613, 1610640355, 1607494599, 1607040811, 1608492986, 6390, 396, 1607734901, 1607350992, 1608992344, 1609903202, 74, 7159, 4698, 7547, 4441, 1607097951, 8479, 8170, 4707, 7516, 1609174124, 1608853150, 8516, 1611424664, 2843, 1607040596, 1953, 1607772081, 1608564004, 2627, 1607217921, 1607118643, 1607314413, 1609887404, 1608335387, 4098, 1607046006, 1608242917, 8918, 1607131299, 9754, 1607262842, 1607273026, 2478, 1607536408, 1607291545, 1608707232, 1609941585, 1608200497, 1610630377, 7711, 1607810287, 9934, 1608850448, 4041, 1609166629, 1608168856, 1607572897, 6831, 1608586814, 1608588581, 2837, 8180, 1608631410, 1607051003, 3830, 4879, 1608920128, 1607019351, 8181, 3473, 1608335906, 1607738757, 1608770486, 7564, 1607495239, 1609983150, 1607397061, 1607696074, 103, 2222]

Unique Train IDs: [4769, 1607928177, 1607269923, 7755, 4598, 1607807806, 1608741452, 3323, 1610110670, 1607133044, 9745, 1607291670, 5245, 4442, 319, 1607133218, 1607010270, 1608587203, 1609256130, 1608582258, 5028, 1609771771, 5229, 3517, 1608595561, 1608048050, 1607410780, 528, 1607134906, 3102, 1607555727, 1609887167, 3985, 7256, 3523, 1607289708, 1609890222, 850, 1608917024, 5047, 1608061691, 4782, 1608062276, 1056, 1611517276, 1607636681, 1607891972, 5571, 1609052616, 1607927243, 2525, 4353, 1610818662, 8640, 1607559849, 6706, 1608624428, 1607968838, 1608672132, 552, 1608537399, 1610381937, 1608607986, 381, 1608589576, 3920, 1608059746, 1609027319, 1607357022, 1607691623,

1609899907, 1608470962, 8791, 1610380419, 3064, 1609473849, 1607712704, 1609887249, 1609888813, 1608588103, 1244, 7279, 1607339125, 1607712682, 8472, 1269, 1607045076, 1607365865, 1846, 191, 1811, 1608702785, 1609049435, 5330, 1607257348, 1609890530, 3278, 1608586899, 1607939718, 2430, 1609893292, 60, 1607270186, 6336, 8650, 1608495626, 1608586953, 2121, 1607295286, 896, 1609889389, 1607560754, 6548, 6580, 1607440988, 1609111416, 1607807159, 8663, 1607129044, 6658, 1607799213, 3933, 1608596696, 1608663032, 1610791060, 1607135820, 1607413039, 1607659758, 1608487726, 4859, 1609142183, 1607276888, 7452, 1607368510, 1607266081, 2623, 1608416516, 2128, 3227, 5881, 6510, 1609166843, 7569, 1607712793, 1608850996, 3273, 1607939838, 9986, 3302, 1607206195, 1609082904, 1607510222, 7612, 1607022963, 1607051040, 1607719324, 1608849324, 1607642639, 1607104225, 705, 1608506424, 1608188073, 8018, 8085, 4755, 1611704179, 1607193886, 7007, 3041, 4001, 1552, 1716, 1608053349, 1608572299, 1608051417, 1607712784, 836, 1607929944, 1607795480, 1608200317, 415, 3662, 1610109929, 2496, 8550, 6868, 1608587385, 1608591490, 7370, 4549, 7505, 1879, 1876, 1608003341]

Duplicated Train IDs: [1607315588, 1607087749, 518, 6280, 1607315467, 1607497867, 1608359052, 8468, 1607124379, 6941, 1607510942, 1607921053, 1611575587, 1608201126, 1607639591, 4521, 1607315498, 1607639340, 1607785646, 1609353141, 1607453496, 1607293881, 1607674683, 5948, 1608683584, 1608087234, 3267, 1607802179, 7109, 838, 9034, 1868, 1608683738, 8284, 95, 1609154655, 1607159523, 1607124838, 1607088999, 7912, 1607802347, 1607764720, 1607785968]

## A.2 Data and code access for reviewers

To access the StudentSADD data, reviewers can use the login information (provided in the submission system) at our project website: `https://emutivo.wpi.edu/data/`

Upon publication, we will grant similar access to other researchers who request our data and agree to the terms of the data licence in Section A.3. This approach is standard for datasets in this domain [32; 34]. This is not a static dataset, as we will continue to add more data features and embedding representations. We have inquired about obtaining a DOI for StudentSADD through our institutional library.

To access the code used to generate the results in this paper, reviewers can navigate to our public github repository: `https://github.com/mltlachac/StudentSADD`

## A.3 StudentSADD Dataset - End User License Agreement

(`https://emutivo.wpi.edu`)

The person in request may download and use this database only after signing and returning this agreement form. By signing this document, the user agrees to the following terms:

**Commercial and academic use**

The database is made available for research purposes only. Any commercial use of this data is forbidden.

**Redistribution**

The user may not distribute the database or parts of it to any third party.

**Publications**

The use of data for illustrative purposes in publications is allowed. Publications include both scientific papers and presentations for scientific/educational purposes. In this case, the identity of the subjects should be protected (no release of identifiable information for subjects).

**Citation**

All publications reporting on research using this database have to acknowledge this by citing the following article:

*ML Tlachac, Ricardo Flores, Miranda Reisch, Rimsha Kayastha, Nina Taurich, Veronica Melican, Connor Bruneau, Hunter Caouette, Ermal Toto, Elke Rundensteiner, "StudentSADD: Mobile De-*

*pression and Suicidal Ideation Screening of College Students during the Coronavirus Pandemic",*
*in Submission at Neural Information Processing System (NeurIPS) 2021 Datasets and Benchmarks*
*Track*

For specific software output that is shared as part of this data, the user agrees to respect the individual software licenses and use the appropriate citations as mentioned in the documentation of the data.

**EULA changes**

Worcester Polytechnic Institute is allowed to change these terms of use at any time. In this case, users will be informed of the changes and will have to sign a new agreement form to keep using the database.

**Warranty**

The database comes without any warranty. In no event shall the provider be held responsible for any loss or damage caused by the use of this data.

---

Name                    Date                    Signature

---

Organization                                    Address

## A.4   Data management plan

**Data description and formats**

The Student Suicidal Ideation and Depression Detection (StudentSADD) dataset contains 245 sessions submitted through a mobile app by 302 students over between August 2020 and January 2021. The data includes PHQ-9, demographics, retrospective phone data, text prompt, audio recordings, location history, and tweets. The audio is stored as WAV files. The remaining data is stored as text in CSV files.

**Data archiving, access and sharing, and data preservation**

Data will be stored in the file systems of Worcester Polytechnic Institute (WPI). The stored data will be protected with disk mirroring, daily backups and other means. Full-time system administrators will monitor the security and availability of these systems. Appropriate access control and other security policies and mechanisms will be put in place to protect the integrity, security, privacy, confidentiality and other rights or requirements. For access by and sharing with the greater research community and general public, research group websites will be used, and personally identifiable information (PII) will be appropriately removed or anonymized from the essential data used for our research. To protect privacy, confidentiality or other rights/requirements, while at the same time ensuring scientific reproducibility and verifiability, the data will be summarized in the forms of intermediate statistics, and made publicly available, or when requested from other researchers.

**Data privacy**

Data will be shared only under rules specified by our IRBs, and only when properly anonymized. The systems on which the data will be stored will have access restricted to the project members via standard filesystem permissions management. The server on which the data is stored is within the University's restricted access datacenter. Files that will be made publicly available will not contain any identifiable location information.

**Policies and provisions for re-use, re-distribution**

Any data collected will be in compliance with the IRB protocols of Worcester Polytechnic Institute. Data gathered for this project may be reused in other, related, research projects conducted by the PIs or graduate students. It is expected that other researchers in machine learning in mental health would be interested in our dataset. Requests for the data would be handled as per Section III. These datasets cannot be used for commercial applications or purposes, or changed and resubmitted without the PIs' permission and are subject to the intellectual property policies of WPI.

**Rights and Obligations**

The Principal Investigators (PIs) will be responsible for the implementation of the Data Management Plan. WPI owns the technology developed at each individual university by each university's employees.

