# OpenReview forum: "StudentSADD: Mobile Depression and Suicidal Ideation Screening of College Students during the Coronavirus Pandemic"
_NeurIPS.cc/2021/Track/Datasets_and_Benchmarks/Round1 — Submitted to NeurIPS 2021 Datasets and Benchmarks Track (Round 1)_

### Official Review · Reviewer_MqFb · 2021-06-30
**Potentially valuable and highly socially relevant dataset, but lack of clarity about what data will be released and privacy concerns.**

**Rating:** 6
**Confidence:** 4

**Strengths:**

- Generally this type of data is very hard to collect. The data is small for ML purposes, but relatively large compared to many existing datasets in this space. It’s also very rich in terms of the data collected.
- Data is collected during a time where concerns related to mental health of students are very relevant given the circumstances.
- Good related work.
- Clear data management plan and end user license agreement.
- Clear description of IRB procedure.



**Weaknesses:**

Weaknesses:
- Throughout the paper I was confused about what data was collected exactly and what data will be released. I wish the authors had included a table listing all the features that were collected, and for each feature: will it be released, not released, only released in processed form (if so, what form). Sentences like "We will share features and embeddings for the data that can not be anonymized, such as voice recordings." are too vague.
- The description of the experimental setup and the models lacked clarity. For example Line 277:  "When screening for depression, the models that used text features (Table 3) only required one feature." This seems to be a result of their feature selection approach that only retained one feature, but this seems odd to me. The exact experimental setup was not clear to me. For example, they only had a train and test set (and no development set), so I'm unsure how parameters were tuned and features were selected.
- The purpose of the research is to determine student willingness to share different modalities (126). This is a very interesting question, but the results remain a bit inconclusive, in part due to technical reasons (e.g. not everyone was able to submit audio even if they wanted to). I wasn't sure why the authors emphasized this so much as one of their main contributions.



**Additional Feedback:**

I wasn't able to download "TextPromptFeatures"

**Clarity:**

Overall the paper is well written.

- I don’t think it’s necessary to provide formulas for accuracy and F1. This space could be used to talk more about the data or experiments.
-  262: 'students were not' -> 'students who were not remote'?
- Table 1: Would be more readable if the digits would align (e.g. according to the '.')
- The title of the paper can be more clear about the context (these are US students).
- The description of the data (statistics etc) could be more clearly separated from the experimental machine learning results.
- Lines 262-263: I’m not too fond of the phrasings  "worst mental health" and "best mental health". The focus of the survey also don’t capture the whole concept of ‘mental health’.

**Correctness:**

The data was carefully collected. The authors were open about possible limitations and technical difficulties that could have affected the data collection. Overall, it seems that a lot of thought was put into the overall data collection process.
I find it hard to interpret the experiments given that some important aspects were unclear (see rest of the review).

**Documentation:**

Clear data management plan and end user license agreement, however the paper lacks clarity about what data exactly was collected and what will be released (and in which form).

**Ethics:**

- Clear about IRB approval (ID provided)
- 183: "extra credit for completing our study". I’m not very comfortable with the idea of professors providing extra credit for students to provide such sensitive information.
- What exactly does the anonymized data contain? Even though explicit references may be left out, the provided information may allow tracing back to an individual--I think this is problematic especially given the combination with the depression screening survey data. For example, looking at the released text prompts, some of them contain quite personal information that I imagine can be traced back to an individual (e.g. my aunt owns a house in…, … .where I go on holiday each year). Furthermore, one could combine such information with e.g. demographic categories (some of these are very infrequent) making it even easier to potentially identify individuals.
- The paper mentions Twitter usernames, but I can’t see how this could be released while maintaining anonymity? (e.g. line 529)
- The paper states that the benefits of screening technology for mental health would outweighs the risk of mis use, but this could be motivated more strongly.
- Twitter has restricted the type of use cases one can use its data for, for example, it states that Twitter data shouldn’t be used to infer information about a user’s health https://developer.twitter.com/en/developer-terms/more-on-restricted-use-cases. How does this apply to the specific use case of this paper?



**Relation To Prior Work:**

Good discussion of prior work.

**Summary And Contributions:**

This paper presents a new dataset with data collected through a mobile application of over 300 US College Students during the pandemic. The data was collected throughout the Fall 2020 semester, the first semester to be fully impacted by the pandemic.

Contributions:
- A new dataset on a highly socially relevant topic. The data is quite rich, containing demographics, phone data, text logs (without content), text prompt, voice recordings, location history, tweets etc. In addition, students filled in a depression screening survey. However, what was collected exactly from each student depended on their consent (only a very small number provided their Twitter account, for example). This is valuable information but also very sensitive (privacy and ethical considerations, see also below).
- Baseline machine learning experiments to screen for depression and suicidal ideation using the collected data.

===

Based on the response, I changed my score from 5 to 6.

---

### Official Review · Reviewer_b1vH · 2021-07-01
**A valuable contribution to the Datasets and Benchmarks Track**

**Rating:** 7
**Confidence:** 4
**Clarity:** Apart from a few missing words/typos,…

**Strengths:**

Especially with the growing concerns around mental health due to the effects of the pandemic, I expect this dataset will be useful and interesting to many researchers in the near future. The dataset is easily accessible, and the potential ethical and social implications explicitly mentioned.

**Weaknesses:**

While the authors have acknowledged and warned against the ethical and social risks, these risks remain. In addition, while the authors explain the types of data within the paper, it would have helped to include examples of the data and the number of rows per type of data - some participants had answered the questionnaire before two questions were added, and others saw some questions at the end of the survey rather than the start, for example.

**Additional Feedback:**

n/a

**Correctness:**

The data types available, and their description, make sense and I would assume are accurate. I cannot comment on the prediction methods using ML due to my unfamiliarity with the methods, although the conclusions from their results appear to be correct.

**Documentation:**

As mentioned earlier, the paper would benefit from an example of the data (1-2 rows). However, apart from this, the dataset is well-documented. The data types in each file is described carefully. The paper includes information on how to access the dataset (even after future updates), a data management plan, and an end user license agreement.

**Ethics:**

The ethical and social risks for any dataset with personal data and particularly mental health data remains for this dataset as well. The models trained on such data can be ran on any person's data that is reachable without their consent, which the authors acknowledge. Besides that, the data could potentially be traced back to the individual who produced the data in the first place; the authors claim that all data has been anonymized and that the types that couldn't (e.g. voice recordings) were excluded. The risks remain for the text data and especially the Twitter data (username and public tweets).

**Relation To Prior Work:**

The work is well-situated in the prior work, which motivates the creation of this dataset quite well.

**Summary And Contributions:**

Thank you to the authors for their work.
The dataset consists of 345 rows with (anonymized) data relating to university students' mental health, their responses to voice/text prompts, and other types of data (e.g. phone call logs). The paper also presents different ML screening methods for depression and suicidal ideation using this data, with varying efficiencies. None of the methods are outstanding, most are in fact pretty unsuccessful at predicting depression and suicidal ideation.

I believe this dataset can be useful to the machine learning community, keeping in mind the risks that the authors mention in the paper. The work is well-motivated with recent research taking into account the pandemic and its global effects. However, I would encourage the authors to acknowledge that the dataset is in the English language (as I assume from their descriptions) and most likely from a majority-American group of students.

I would recommend accepting this paper to the Datasets and Benchmarks track.

========
Update (20 July 2021): I am happy with the authors' responses and changes. Thank you to all the authors and reviewers for their work.

---

### Official Review · Reviewer_1tcT · 2021-07-06
**The paper is well written but the data collection process introduces biases in the dataset (modified: 19 Jul 2021)**

**Rating:** 5
**Confidence:** 4
**Clarity:** The paper is will written, logically …

**Strengths:**

The authors present a new dataset for evaluating depression and suicidal ideation in college students during the covid-19 pandemic. The contribution is novel in terms of the time of collection, while student were remote or under strict restrictions on campus. It evaluates a range of modalities for their usefulness in screening for depression and suicidal ideation, as well as the willingness of students to share. It is an alternative to traditional questionnaires used for the same purpose. This study is particularly important due to the increase in suicide and depression rates, particularly amongst young adults. The authors use a variety of machine learning and deep learning models to calculate the accuracy for both depression and suicidal ideation.

**Weaknesses:**

Even though the authors look at a variety of data modalities, these are not equally available to all students. The app is only available for Android, which limits its use only to students that have Android phones and also sets selection biases associated with groups that might prefer a certain phone rather than another. Phone-related data are only available to collect from students that use the app (rather than the website), which significantly limits the availability of this data modality. In addition, technical problems with audio collection mean that the evaluation of willingness to share might not be accurate for this modality.

The order in which participants are presented with the option to submit data modalities also sets biases. For example, almost all students submitted demographic data after the authors put them directly after the PHQ-9 page (rather than at the end). This is likely the same reason many students did not reach the Twitter page. The authors could have switched up the order for different participants (especially since many participants did not have a Twitter account, which further limited the availability of this modality), in order to obtain a more objective evaluation of willingness to share. The authors end up using only the audio and text modalities.

**Additional Feedback:**

If this is not a benchmark, the authors should remove the word benchmark from line 65. If this is also a benchmark, the methodology should be explained in more detail and the choices the authors have made in terms of the modeling should be significantly more justified.

The authors detail their hyperparameter choices only for one model. They also mention that  the maximum number of trees used for training tree-based models was chosen to be 3 to prevent overfitting. This statement does not seem sufficiently justified.

It's not clear why the authors choose 1-10 principal components or features for training.

The authors should explain how they normalized the data.

(modified: 19 Jul 2021)

**Correctness:**

The claims made in the submission are overall correct and the dataset is structured in a sound way. The dataset includes biases due to reasons outlined above.

**Documentation:**

The dataset collection process is described in detail. Part of the dataset is available in the form of feature embeddings to preserve privacy. The dataset will keep being updated and enriched. The data is available by the authors upon request.

**Ethics:**

The ethical and social implications are the use of audio and text recordings to screen for depression and suicidal ideation without permission from the data owner.


**Relation To Prior Work:**

Yes, previous contributions are clearly discussed and the differences with the current paper clearly outlined.

**Summary And Contributions:**

The submission presents a new dataset for screening for depression and suicide ideation in college population during the covid-19 pandemic. Its main contributions are:
- A new, less intrusive way to collect data with a focus on college students
- A dataset including more students and a richer variety of almost instantaneously obtainable modalities
- An evaluation of willingness to share these modalities
- An evaluation of how useful these modalities are for screening depression and suicidal ideation using machine learning and deep learning models
- The dataset was collected while students were either virtual or facing strict restrictions on campus

---

### Decision · Program_Chairs · 2021-07-26

**Decision:**

Reject

**Comment:**

This paper presents a new dataset for automated screening for depression and suicidal ideation in college populations. One of the key novelties is the inclusion of multiple modalities (including twitter data and text responses to prompts).

The reviewers agree that the dataset is novel and could be an important resource, particularly during the time of COVID-19.

However, reviews are split overall on the suitability of the current draft of this paper for publication. There are clear concerns regarding the validity and meaningfulness of the results, e.g. that more students using the web app (vs phone app) to submit data indicates a *preference* for the web app rather than a result of popularity of android phones, the method of survey distribution, etc. Ethical concerns have also been raised by reviewers, although the authors do discuss some of the limitations and ethical risks in the draft.

I recommend the authors revise the manuscript with greater attention to the biases in the dataset, include additional analysis of the dataset statistics, and include a discussion of the limitations in light of the results. I also encourage the authors to re-submit the revised and strengthened manuscript to the second round of the track.